# Applications of Wireless Sensor Networks: An Up-to-Date Survey

**Dionisis Kandris** [1,*] , **Christos Nakas** [1] , **Dimitrios Vomvas** [1] **and Grigorios Koulouras** [2,3]

[1] microSENSES Research Laboratory, Department of Electrical and Electronic Engineering, Faculty of Engineering, University of West Attica, GR-12241 Athens, Greece; mscres-2@uniwa.gr (C.N.); mscee17009@teiath.gr (D.V.)

[2] TelSiP Research Laboratory, Department of Electrical and Electronic Engineering, School of Engineering, University of West Attica, GR-12241 Athens, Greece; gregkoul@uniwa.gr

[3] Hellenic Telecommunications and Post Commission, GR-15125 Athens, Greece

[*] Correspondence: dkandris@uniwa.gr

**Abstract:** Wireless Sensor Networks are considered to be among the most rapidly evolving technological domains thanks to the numerous benefits that their usage provides. As a result, from their first appearance until the present day, Wireless Sensor Networks have had a continuously growing range of applications. The purpose of this article is to provide an up-to-date presentation of both traditional and most recent applications of Wireless Sensor Networks and hopefully not only enable the comprehension of this scientific area but also facilitate the perception of novel applications. In order to achieve this goal, the main categories of applications of Wireless Sensor Networks are identified, and characteristic examples of them are studied. Their particular characteristics are explained, while their pros and cons are denoted. Next, a discussion on certain considerations that are related with each one of these specific categories takes place. Finally, concluding remarks are drawn.

**Keywords:** wireless sensors; wireless sensor networks

---

## 1. Introduction

A Wireless Sensor Network (WSN) is a group of spatially dispersed sensor nodes, which are interconnected by using wireless communication [1]. As seen in Figure 1, a sensor node, also called mote, is an electronic device which consists of a processor along with a storage unit, a transceiver module, a single sensor or multiple sensors, along with an analog-to-digital converter (ADC), and a power source, which normally is a battery. It may optionally include a positioning unit and/or a mobilization unit.

A sensor node uses its sensor(s) in order to measure the fluctuation of current conditions in its adjacent environment. These measurements are converted, via the ADC unit, into relative electric signals which are processed via the node's processor. Via its transceiver, the node can wirelessly transmit the data produced by its processor to other nodes or/and to a selected sink point, referred to as the Base Station.

As illustrated in Figure 2, the Base Station, by using the data transmitted to itself, is able to both perform supervisory control over the WSN it belongs to and transmit the related information to human users or/and other networks [2].

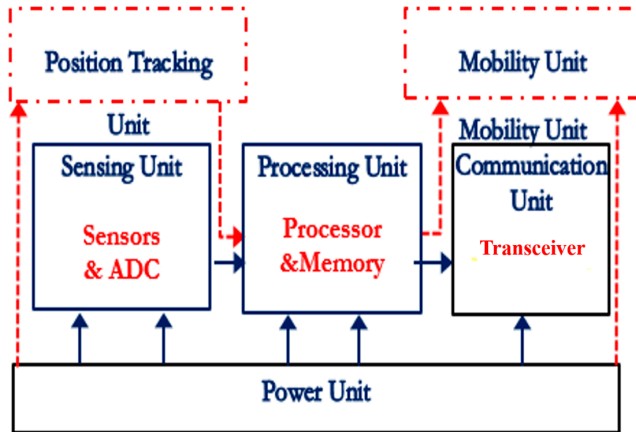

**Figure 1.** The typical architecture of a sensor node used in Wireless Sensor Networks (WSNs).

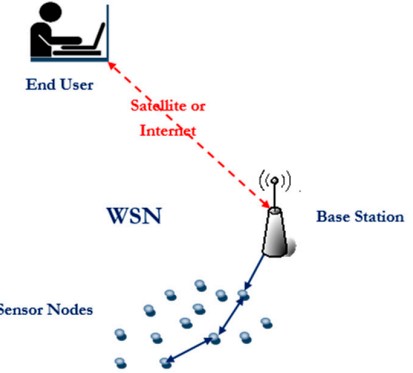

**Figure 2.** The typical architecture of a WSN.

The collaborative use of a sufficient quantity of such sensor nodes, enables a WSN to perform simultaneous data acquirement of ambient information at several points of interest positioned over wide areas. The inexpensive production of sensor nodes of this kind, which despite their relatively small size, have exceptionally advanced sensing, processing, and communication abilities has become feasible due to continuous technological advances. For this reason, although WSNs were initially used mainly for military purposes, nowadays they support an ever-growing range of applications of different types [3].

The aim of this paper is to study characteristic examples of different existing applications of WSNs, both widely used and novel ones, and examine various issues of interest related to this topic. Specifically, in Section 2 applications of WSNs are classified according to their nature into corresponding categories and their specific features are examined, through the presentation of indicative examples. Specific concerns arising from this study are discussed in Section 3. Finally, concluding remarks are drawn in Section 4.

## 2. Main WSN Applications

Various applications of WSNs are currently either already in mature use or still in infant stages of development. In this paper, WSN applications are classified according to the nature of their use into six main categories which, as illustrated in Figure 3, namely are: military, health, environmental, flora and fauna, industrial, and urban.

In each category, various subcategories are considered. In what follows in this section, the nature of each one of these categories and subcategories is explained. Additionally, through the indicative examination of characteristic examples of them, their particular features are explained, while their benefits and problems are denoted.

Moreover, various methodologies and technical means that are used in these applications either for sensing or for processing purposes are discussed, while similarities and dissimilarities existing among them are identified.

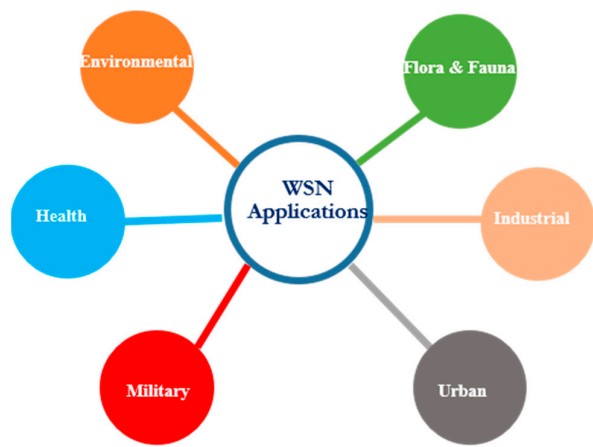

**Figure 3.** Overview of the most popular categories of applications of WSNs.

## 2.1. Military Applications

The military domain is not only the first field of human activity that used WSNs but it is also considered to have motivated the initiation of sensor network research. Smart Dust [4] is a typical example of these initial research efforts, which were performed in the late 90 s in order to develop sensor nodes which despite their very small size would be capable of accomplishing spying activities.

The technological advances achieved since then made WSNs capable of supporting various operations [5]. In Figure 4, the main subcategories of the military applications of WSNs which namely are battlefield surveillance, combat monitoring, and intruder detection, are illustrated along with the types of sensors that are most commonly used in them.

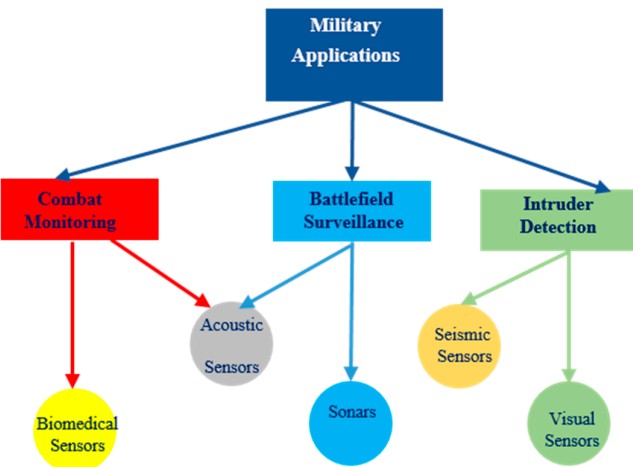

**Figure 4.** The subcategories of the military applications of WSNs and the types of sensors they use.

Specifically, Chemical, Biological, Radiological, Nuclear and Explosive (CBRNE), and Toxic Industrial Material (TIM) sensors may be used to detect the presence of such substances. To detect intrusion, WSNs may use infrared, photoelectric, laser, acoustic, and vibration sensors. Similarly, RAdio Detection And Ranging (RADAR), LIght Detection And Ranging (LIDAR), LAser Detection And Ranging (LADAR) and ultrasonic sensors are used by nodes in WSNs in order to detect the distance from objects of interest. Likewise, LADAR and infrared sensors are used for imaging purposes.

Additionally, the flexibility that WSNs have in their structure enables them to adapt to various requirements. For instance, in battlefield operations large-scale WSNs consisting of many thousands of nodes, which are non-manually deployed, are used. In urban warfare and force protection operations, WSNs used consist of hundreds of manually deployed nodes. In other-than-war operations all scales of WSNs and deployment methods are used [5].

WSNs that have been developed for battlefield surveillance, combat monitoring, and intruder detection are examined in the following subsection.

### 2.1.1. Battlefield Surveillance

In applications of this type, the sensor nodes of the WSN may be deployed on a battlefield nearby of the paths that enemy forces may use. The main advantage provided is that the WSN not only can be spontaneously positioned but also can function, without need for continuous attendance and maintenance. The terrain of the battlefield in most cases is absolutely variable. This plays an important role for both the coverage and the energy consumption of the sensor nodes. Some of these applications are presented below.

In [6], the use of WSN technology for ground surveillance is studied. Specifically, the authors propose a system that consists of low-cost common nodes which are capable of sensing magnetic and acoustic signals produced by various moving target objects. The system aims to detect and categorize various targets, such as vehicles and troop movements, based on the spatial differences of the signal strength detected by the sensors. Figure 5 illustrates the architecture of this system.

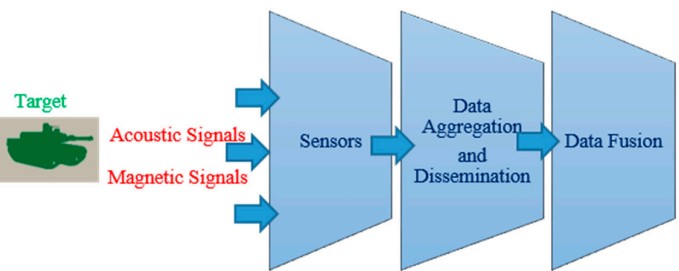

**Figure 5.** Illustration of the system architecture proposed in [6] for target tracking.

In [7] a submarine detection system for Anti-Submarine Warfare (ASW) is presented. The system consists of inexpensive multi-sensing units that combine both active and passive sonars. The system can be scaled to a large number of sensors, which are deployed in littoral waters according to a specific pattern and ocean depth. These sensing units utilize their passive sonar to detect a diesel-electric submarine and their active sonar to confirm their target. The unit that has confirmed a target, notifies its neighboring sensing units by using an alarm signal that contains its ID code. Whenever the units send multiple alarm signals within a predefined period of time, an alert is triggered. Furthermore, the system can acquire low False Alarm Rate (FAR) due to the very low range of the active sonar (50 m) that solves the acoustic multipath problem of the conventional sonobuoys.

### 2.1.2. Combat Monitoring

Within a battlefield the firing of guns, mortars artillery, and other weaponry creates sound, heat, and vibrations. This information can be recorded with the use of WSNs and provides an expectation of the location of the enemy. This type of application is described below.

In [8] the use of acoustic sensor arrays suspended below tethered aerostats to detect and localize moving vehicles, transient signals from mortars, artillery, small arms fire, and locate their source is presented.

The specific detection system can be used in conjunction with an acoustic vector sensor, to amplify the possibility of locating the threat using the shockwave created by the supersonic bullet and the muzzle blast created by the gun [9].

The authors of [10] describe a system consisting of interconnected Body Sensor Networks (BSNs), a sub-family of WSNs, for real-time health monitoring of soldiers. A key component of this system is a BSN that integrates various physiological and biomedical sensors. These sensors are an accelerometer, an EEG simulator, and a $SpO_2$ sensor, which can be embedded within an advanced combat helmet worn by each soldier. They monitor the various information of health status in real time, such as blood pressure, oxygen saturation, and heart rate. By utilizing the data collected, various methods can be applied in order to train soldiers more efficiently and prepare them more adequately for future engagements.

### 2.1.3. Intruder Detection

The knowledge of the location of the enemy is considered to be one of the most critical pieces of information during military operations. Whichever side in a conflict has this knowledge is one step ahead and closer to victory. With the utilization of WSNs within a battlefield, intruders can be detected in good time to prevent loss of supplies or territory.

An intruder can be detected in various ways. The authors of [11] developed a system that can recognize intruders by utilizing Unattended Acoustic and Seismic Sensors to record vibrations and sounds of typical soldier actions. To detect an intruder, with the aid of these sensors, typical military activities, such as equipment handling, walking, rifle loading, etc., have been measured and recorded in a controlled environment and in the field. Although the distance of detection may be small, the information provided by the network can be used to locate an intruder in difficult terrains with large vegetation where visibility is limited.

In [12] another application of WSNs, developed to detect intruders, is introduced. The authors described a system with dense deployment of a large number of miniaturized wireless sensors, which are small, low cost, and consume low energy. This system utilizes acoustic and seismic sensors to provide detection information to the user and tracking and visual sensors in order to lower the false alarm rate.

### 2.2. Health Applications

In the health domain, WSNs utilize advanced medical sensors to monitor patients within a healthcare facility, as a hospital or within their home, as well as to provide real time monitoring of patient's vitals by utilizing wearable hardware. In Figure 6, the main subcategories of health applications of WSNs namely patient wearable monitoring, home assisting systems, and hospital patient monitoring are illustrated along with the types of sensors that are most commonly used in them. WSNs that have been developed for these types of health applications are examined in the following subsection.

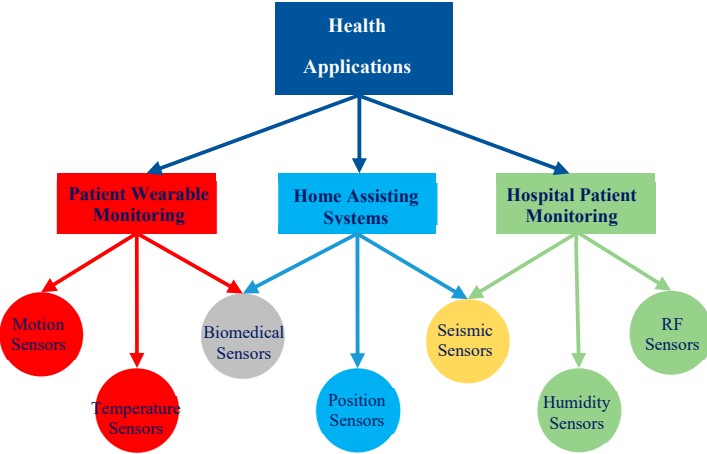

**Figure 6.** The subcategories of the health applications of WSNs and the sensors used in them.

### 2.2.1. Patient Wearable Monitoring

Health monitoring applications can be combined with wearable hardware with embedded biomedical sensors that provide the patient's health status in a remote environment or within a healthcare facility.

A healthcare solution of this type, is examined in [13]. It uses real-time sensors incorporated in smartphones along with a barcode system to provide personalized medicine care assistance. The mechanism developed, performs Electrocardiogram (ECG) monitoring in real time. Also, the monitoring of the blood glucose level, blood pressure, and several kinds of diagnostics could be possible too, by using real-time sensors. This system developed is illustrated in Figure 7.

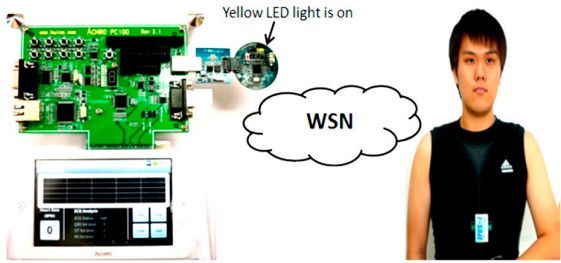

**Figure 7.** An illustration of the real time electrocardiogram (ECG) monitoring environment [13].

An alert portable tele-medical monitor system (AMON) is proposed in [14], aiming to provide continuous monitoring for high-risk cardiac/respiratory patients. The system collects multiple vital signs, detects multi-parameter medical emergencies and is connected to a cellular telemedicine center (TMC). This system uses a wrist worn device, to monitor vital parameters of patients in order to provide an integrated picture of their health condition.

### 2.2.2. Home Assisting Systems

The authors of [15] presented a homecare monitoring platform with internet remote connection to assisted people and their environment. This platform can monitor and diagnose patients remotely, in real time in their home environment by the utilization of wearable or even surgically inserted biosensors. A hybrid model of the platform is proposed, with various levels in terms of functionality, combining fixed and mobile nodes. These levels range from simple data acquisition of the assisted person to primary care and emergency nodes, up to the communication and coordination with an appointed help center as a hospital.

During their research, the authors of [16] investigated real-time sensors for the diagnosis of cardiac patients by using smartphone and wearable sensors. The authors introduced this application to cater a remote real time monitoring for severe cardiac patients unable to attend a routine checkup. One of the main reasons behind this technology is to commercialize it for the benefit of those patients who are not financially sound. Furthermore, the system was developed in such a manner that any kind of crisis phase is dealt with in terms of alerting messages that are automatically sent to the doctor. An overview of the operation of this system is shown in Figure 8.

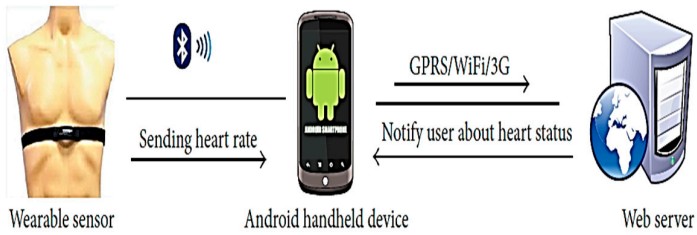

**Figure 8.** The system configuration of the remote patient monitoring application [16].

### 2.2.3. Hospital Patient Monitoring (or Hospitalization)

Within healthcare facilities, such as a hospital, WSNs systems can be integrated to provide real time patient monitoring and emergency alerting for a more precise and quick response. A typical example of an application of this kind is presented in [17]. Specifically, the authors described a WSN application where wireless sensors are placed within the emergency rooms of John Hopkins hospital to monitor in real time the blood oxygen and heart rate of the patients. The researchers collected the performance statistics of the network and despite the difficulty of the hospital environment due to interference and radio noise, the application of WSNs can improve the operation of a healthcare facility. The device developed is depicted in Figure 9.

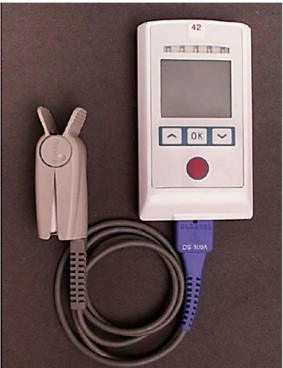

**Figure 9.** A monitor that can both display and transmit vital signs [17].

### 2.3. Environmental Applications

Environmental applications that demand continuous monitoring of ambient conditions at hostile and remote areas can be improved with the utilization of WSNs. In Figure 10, the main subcategories of environmental applications of WSNs, namely water monitoring, air monitoring, and emergency alerting, are depicted along with the types of sensors that are typically used in them. WSNs that have been developed for these types of environmental applications are studied in the following subsection.

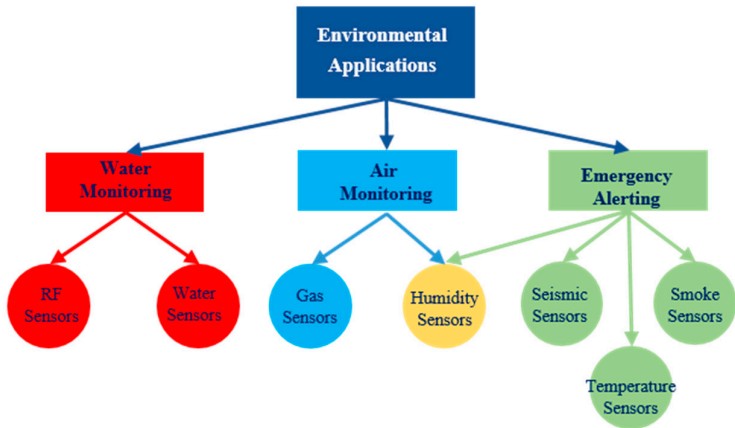

**Figure 10.** The subcategories of the environmental WSN applications and the types of the sensors used in them.

### 2.3.1. Water Monitoring

Water, either for drinking or oceanic is an important factor in human lives, therefore the monitoring of water has a great academic interest for researchers as described below.

The researchers in [18] designed a WSN application to evaluate the quality of fresh drinkable water. They designed a Cyber physical system (CPS) called PipeSense, which is an in-pipe system for

water monitoring that utilizes RFID (Radio Frequency Identification) based WSN. The network can provide information about water demand or water quality and various repair information such as weak spots or pipe leakage. The in-pipe RFID sensors collect information from the system and send them to the data servers, where algorithms provide decision support.

A WSN application for marine environment monitoring is presented in [19]. In order to prevent damage to the flora and fauna of a fish farm from feed and fecal waste, the authors designed an Underwater WSN (UWSN) with ground based wireless sensor nodes capable of monitoring the pollution of the farm. The sensor nodes are mobile in a limited space in order to measure a greater area.

### 2.3.2. Air Monitoring

Air is a vital element for human lives and nowadays the air pollution of the atmosphere is a result of many modern human activities. WSNs can be utilized for air quality monitoring in occupied regions in order to prevent dangerous diseases and contaminations or risk the health of people.

A characteristic example of an application of this kind is presented in [20], where an air quality monitoring application assisted by WSNs, called WSN-AQMS, is proposed. The specific system combines gas sensors along with Libelium waspmotes [21] to measure air quality parameters of gases such as ozone, CO, and $NO_2$. The waspmotes monitor in real time the air quality and utilize the Zigbee protocol for data communication. The authors further introduced the Clustering Protocol for Air Sensor network (CPAS) in order to support the operation of this system

### 2.3.3. Emergency Alerting

Proactive monitoring of the causes of natural disasters, can help to avoid these disasters or/and lower their cost. WSNs can be utilized for monitoring common disastrous causes in real time to provide proactive alerts in order to lower damage or even prevent disaster. Typical examples described in the rest of this subsection, are related to the monitoring of seismic activity, volcanic activity, forest fires, and tsunamis.

#### Seismic Activity Monitoring

Earthquakes can cause enormous damage to an occupied region where they take place. WSNs can be utilized to monitor seismic activity in real time in order to take precautionary measures and enable the authorities to act in advance. A real time seismic activity monitoring system is presented.

In [22], the authors designed a warning system for earthquakes in order to increase the time before an earthquake so as to take precaution measures. The authors deployed a WSN in the island of Mauritius, which has high seismic activity. The system monitors seismic activity by utilizing primary waves (P-waves) and estimates local velocity and the hypocenter's location according to time delays in the arrival of the P-waves at the sensors.

#### Volcanic Activity Monitoring

Volcanoes can cause enormous damage to nearby towns or cities when they are activated. Before a volcano erupts, there are many signs that a WSN system can measure, proactively, in order to inform nearby citizens about the eruption. When such a system is applied, citizens can protect their families and belongings by transporting them outside of the area of the eruption, preventing further damage. Below is an example of such a system.

In [23], a WSN based system to monitor volcanic activity components is proposed. The system is low cost, flexible, and easy to deploy and to maintain for remote locations. The users of the system can choose GPS data synchronization when the sensor nodes have signal reception, or a specific algorithm when they have not, to collect accurate timestamps of each sample. Pieces of the equipment used, are shown in Figure 11.

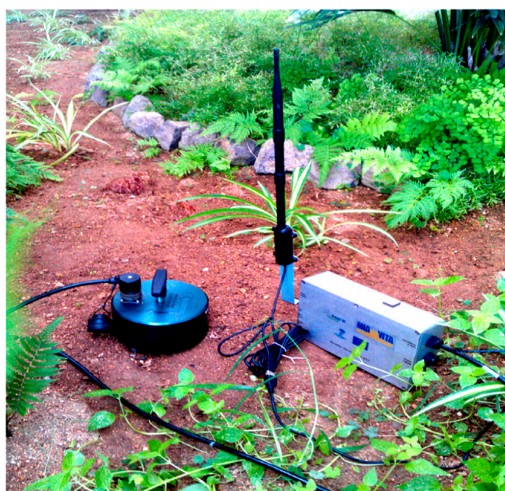

**Figure 11.** A depiction of a geophone sensor with a wireless antenna deployed in the field [23].

Forest Fire Prevention

Forest fires can destroy both animals and vegetation, thus resulting in huge ecological disasters. In order to prevent such disastrous events WSNs can be deployed within forests and monitor in real time related parameters in order to assist in forest fire prevention. One example of this type of WSN application is presented below.

Due to the rapid climate change taking place, the necessity for proper forest fire prevention means is extremely pressing. Such an example is described in [24], which combines traditional methods (patrols, watchtowers, satellite imaging) to a WSN array monitoring system using ZigBee protocol [25]. Related parameters as temperature, humidity, etc. are monitored in real time and sent to a monitoring center in order to be analyzed. Then the system can make quick estimations of fire danger and inform the authorities.

Tsunami Detection

Another natural disaster is the tsunami waves. It is crucial for coastal regions to be informed early of such a catastrophe. WSNs can be utilized for real time monitoring enabling the authorities to act proactively. Below a Tsunami detection WSN application is presented.

In [26], a tsunami detection system which utilizes a WSN with underwater sensor nodes deployed in coastal regions is proposed. In order to inform proactively the authorities, the system utilizes a lexical resource messaging system, called SentiWordNet, which is able to provide information extracted from the sensor messages.

*2.4. Flora and Fauna Applications*

Both flora and fauna domains are vital for every country. In Figure 12, the main subcategories of flora and fauna applications of WSNs which namely are greenhouse monitoring, crop monitoring, and livestock farming, are illustrated along with the types of sensors that are most commonly used in them.

WSNs that have been developed for these types of flora and fauna applications are examined in the following subsection.

2.4.1. Greenhouse Monitoring

An important sector of the agriculture domain involves greenhouses. Within them many crops can be grown to provide sustainable food while climate crops can be harvested all year round if certain conditions are applied within the greenhouse. Therefore, WSNs can be applied in greenhouse monitoring and control to improve their operation. Below are some examples of these applications

In [27], a system of this type, called the Agricultural Environment Monitoring System (AEMS), is presented. It is an inexpensive and easy to apply system that can collect and monitor data related to crop growth, inside or outside a greenhouse via WSN sensors and CCTV cameras. The system gathers vital environmental parameters such as temperature, light intensity, humidity, air pressure, rainfall level, pH, and electrical conductivity (EC).

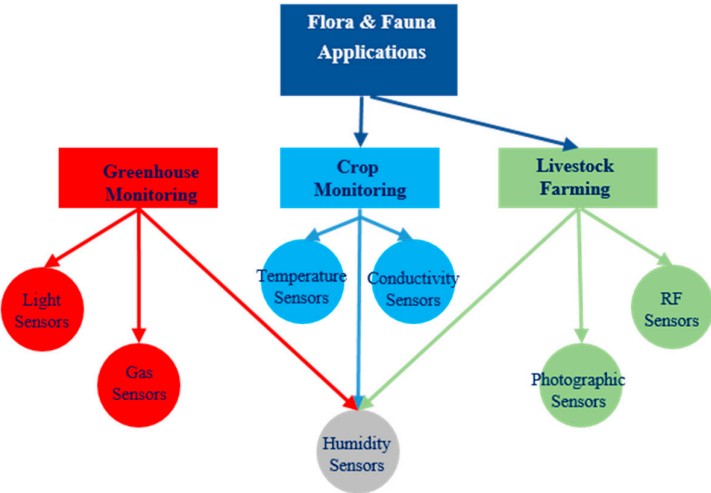

**Figure 12.** The subcategories of flora and fauna applications of WSNs and the types of the sensors used in them.

A relevant system developed for greenhouses which has energy management and indoor climate control capabilities, is introduced in [28]. The system monitors vital greenhouse parameters such as indoor luminance, temperature, and relative humidity, via sensor nodes. The indoor climate control is possible by the utilization of two fuzzy logic controllers, P (Proportional) and PD (Proportional-Derivative) that use the desired indoor climatic set-points. Furthermore, the system utilizes output actuations of heating units, motor-controlled windows and shading curtains, artificial lighting, etc. in order to achieve more precise greenhouse control.

### 2.4.2. Crop Monitoring

Within the agriculture domain, the preservation of the crops plays a vital role. In order to provide a better environment for the crops, various WSNs applications can be implemented. Crop irrigation and fertilization are some examples of the applications that have been designed as described below.

Hidro Bus is a system described in [29]. It is a remote controlled, automatic irrigation system for large areas of land applied at Jumilla (Murcia, Spain). The system divides the deployment area into seven sub-regions, where each has a control sector for monitoring and controlling. The control sectors communicate with each other and the central controller via WLAN network.

An automated fertilizer applicator system is built in [30], which consists of three modules, the input, the decision support, and the output modules. The input module can provide to the Decision Support System (DSS) module, GPS and real time sensor data by utilizing bluetooth technology. With these data the DSS is able to calculate quantity, application rate, and the spread pattern of the fertilizer that the output module will provide to the crops.

The authors of [31] developed and integrated an optimal fertilization decision support system using a wireless sensor LAN, IEEE 802.11 protocol (Wi-Fi) and a GIS analysis server. Sensor nodes were used to acquire vital data in real time such as soil moisture, conductivity, temperature, etc. Also, the system used B/S a (browser/server) structure mode to provide high interactivity. Also, a GIS analysis server was used to interpolate the data from small experimental plots to larger plots to exploit data

reduction for energy conservation. An overview of the overall system architecture is illustrated in Figure 13.

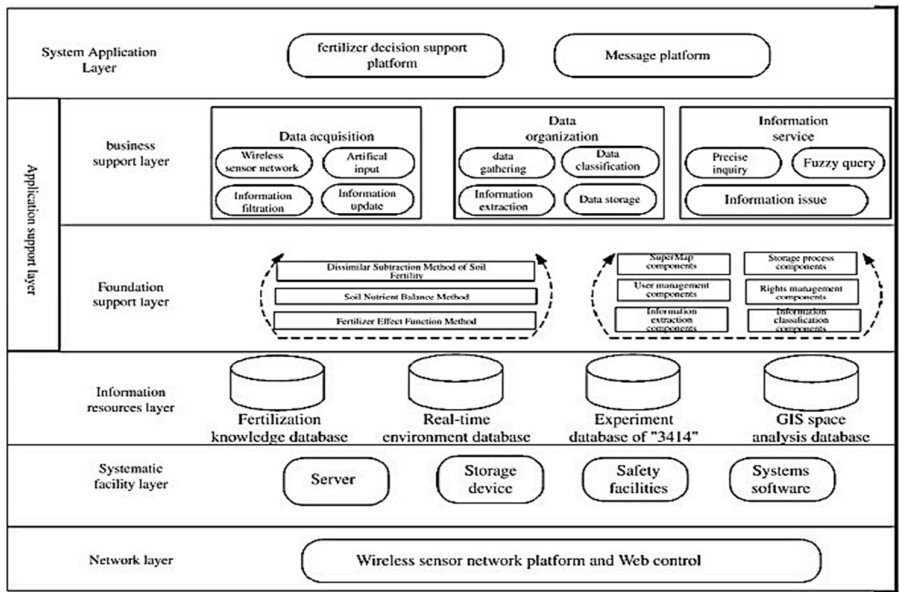

**Figure 13.** A structure diagram of the integrated optimal fertilization decision support system [31].

Similarly, in [32] a scheme based on the collaboration of an integrated system for an automated irrigation management with an advanced novel routing protocol for WSNs, was proposed. This system aims at efficiently managing water supply in cultivated fields in an automated way. The system takes into consideration the historical data and the change in the climate values to calculate the quantity of water that is needed for irrigation.

### 2.4.3. Livestock Farming

Livestock farming is a main sector of the fauna domain. WSNs can be applied in various tasks such as livestock monitoring. Below we have collected some examples of these applications.

In [33], a sensor-based system developed to analyze the behavior of livestock animals and support their farming was proposed. Green pastures are used by cattle for grazing, so grass growth was analyzed through photographic sensors, so that animals can be moved towards them. Their major objective of work was to design rugged hardware that could be used outdoors for modeling both individual and herd behavior of animals. Although, specially designed sensors to monitor animal behavior, such as sleeping, grazing, and ruminating, are used, cattle monitoring still poses several challenges like radio attenuation caused due to factors such as animal body, mobility etc.

In [34], an inexpensive low power collar designed to assist livestock farming is proposed. The system developed uses a solar power relay router and two antennas placed so that the collar radio coverage is optimized. Also, a novel protocol, named Implicit Routing Protocol (IRP), was proposed in order to cope with the packet losses, which are caused due to the mobility of the animals.

In [35], the authors describe a system based on RFID and WSN technologies, which can be applied in modern farm large-scale management systems. The proposed system can identify and track all animals (i.e., pigs) that are present within the farm, and monitor their health as well as the environmental conditions. According to the authors this system, by the aforementioned parameters, can be used to prevent diseases, determine environment pollution, and inform on food safety issues.

### 2.5. Industrial Applications

WSNs can be applied in various industrial applications to solve many related problems. In Figure 14, the main subcategories of industrial applications of WSNs namely logistics, robotics, and

machinery health monitoring are illustrated. These specific categories of applications are studied in the rest of this subsection.

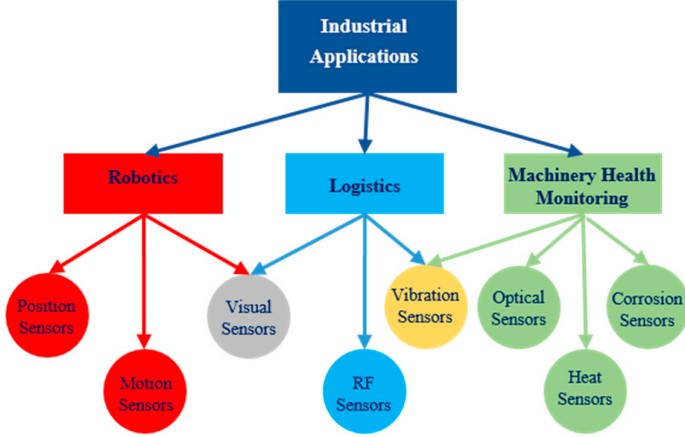

**Figure 14.** The subcategories of the industrial applications of WSNs and the types of the sensors used in them.

### 2.5.1. Logistics

The domain of logistics is an area of interest where WSNs can be applied, because many logistics systems need real time monitoring of various environmental parameters and better handling of packages. These requirements can be fulfilled by combining the logistics systems with WSNs. Some types of these applications are described below.

The transport logistics sector requires low cost and high-quality during deliveries. In [36], the development and deployment of a WSN based system for monitoring transportation conditions, such as temperature and humidity, within a cargo container travelling via both a trans-Atlantic cargo vessel and a lorry is described. The main idea for the deployment of the monitoring system is depicted in Figure 15. It is shown that the use of a system of this kind, can increase quality by providing better supervision and lower the cost by reducing losses during transportation.

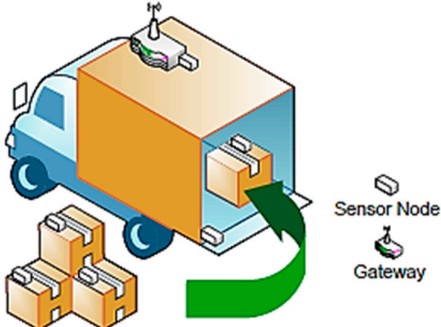

**Figure 15.** Depiction of the basic configuration of a WSN based system for transport logistics applications [36].

The application of WSNs in Cold Chain Logistics (a continuous temperature-controlled supply chain) can greatly improve the monitoring and management of these chains [37]. WSNs are suitable for real time monitoring of many environmental parameters and provide accurate data collection that meets the demand of Cold Chain Logistics. The researchers used a Zigbee ad hoc network model to build the system's framework. By using fuzzy control decision, the environmental parameters are maintained in a stable range and with Maximum Similarities Multiple Characteristic Recognition (MSMCR) the safety of cold-chain food is ensured.

Another application of WSNs developed for Cold Chain Logistics is studied in [38]. This system focuses only on aquatic products and their transportation. In order to develop such a system, the authors proposed a WSN integrated with Compressed Sending (CS), in order to combat heavy data traffic from the real time sensor data transmission. Also, CS provides a low complexity approximation which assists storage, transmission, processing, and meets the recourse constraints of WSNs.

A WSN application in logistics that utilizes GPS technology is described in [39]. This system monitors in real-time the status of the goods and has embedded a terminal, which is used to locate the goods and a cloud services platform, which is used to identify the recipient.

### 2.5.2. Robotics

Nowadays there are many applications that combine WSNs and robots. Robots can cooperate and combat some of the major problems of WSNs, such as sensor node mobility, node redeployment, travelling salesman, etc. Typical WSN applications of this kind are presented below.

A robotic navigation method proposed in [40] provides road maps for the robot to traverse. It uses a WSN with sensors, designed to provide sophisticated maps of their sensing areas. Specifically, each sensor constructs a map, based on the traversable area sensed. Then, all sensor maps are combined to create one large map. Once the road maps are generated, the sensors are used to sense areas of interest for the robot to travel to. The robot then considers all possible roads to take and selects the most efficient path available in the network. If an area becomes hazardous for the robot, the network can reconfigure the road map, and remove this hazardous area from the list of available paths.

One of the most difficult tasks concerning WSNs is the maintenance of a projected network. A combination of WSNs and robotics was researched in [41], where a robotic network servicing system, named Randomized Robot assisted Relocation of Static Sensors (R3S2) was developed. In R3S2, robots move around a network which is contained within a virtual grid. The robot moves to the least recently visited grid point, searching for sensing holes in the network. When an area that is not being covered by a sensor is detected, the robot will find a node which has overlapping coverage with other nodes and move it to the uncovered area. In addition, if the robot discovers redundant sensors, it will move the nodes to cover a greater area.

A different approach is proposed in [42], where a mobile robot is used to transfer data from a widespread network between nodes that are out of reach from one another for various reasons. This is the so-called travelling salesman problem (TSP)—within a WSN. By having a robot traveling among nodes which are out of each other's wireless communication range, it allows the network to be widespread while also saving power by using the robot for data muling.

### 2.5.3. Machinery Health Monitoring

The objective of machinery health monitoring is to examine the performance of various types of technical equipment and to either detect or predict the occurrence of faults that are obstructive or even catastrophic for their operation.

In [43], a WSN is developed in order to perform energy usage evaluation and condition monitoring for electric machines. The motor efficiency and health condition are estimated non- intrusively by using wireless nodes that monitor the motor terminal quantities (i.e., line voltages, line currents, and temperature, with no interference with the operation of the electric machines.

In [44], a WSN based monitoring system of oil and gas pipelines, named REMONG, is proposed. In order to detect the existence of leakages, the system uses wireless sensor nodes which monitor the pressure and temperature of the pipeline fluid at several points of interest on the pipelines which are stretched over large geographical areas.

In [45] a WSN designed to enhance safety in industrial machinery consisting of a main vehicle and an attached trailer is proposed. A 3D accelerometer and a 3D magnetometer, incorporated in a sensor system device, monitor the trailer operating conditions and the corresponding data are wirelessly transmitted to a processing unit which executes a stability control algorithm. A vibrational energy

harvesting system developed, converts kinetic energy from trailer natural vibrations to electrical energy for the system power supply.

### 2.6. Urban Applications

The variety of sensing abilities offered by WSNs also provides an opportunity to gain an unprecedented level of information about a target area, be it a room, a building, or outdoors. WSNs are indeed a tool to measure the spatial and temporal features of any phenomena within an urban environment, providing a limitless number of applications. The most popular applications of WSNs in the urban domain are related to smart homes, smart cities, transportation systems, and structural health monitoring, as depicted in Figure 16 and further described in the rest of this subsection.

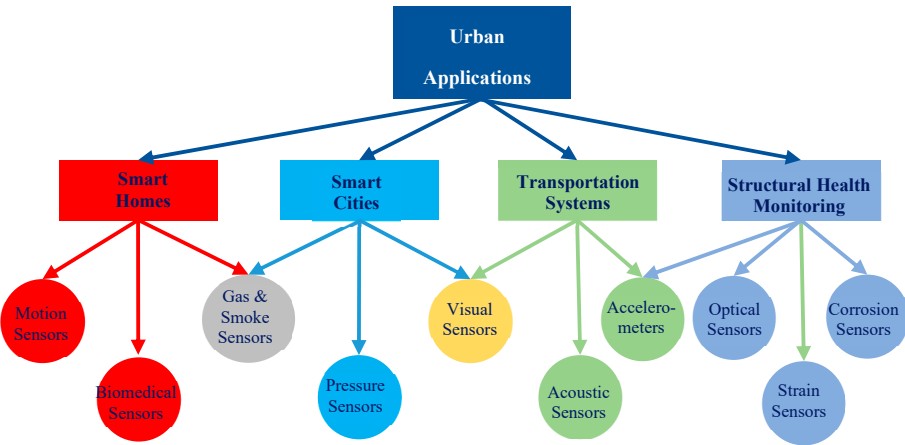

**Figure 16.** The subcategories of the urban applications of WSNs and the types of the sensors used in them.

### 2.6.1. Smart Cities

In large cities monitoring of various parameters is crucial for the optimal life of the citizens. WSNs have a huge amount of applications to provide real time data to the authorities for the optimal function of a city. Specifically, increased transportation of people creates problems and time wastage, when a large number of vehicles is heading towards common destinations. WSNs can be utilized for monitoring in order to reduce traffic, indicate car parking spots, etc. [46].

In [47] a WSN-based intelligent car parking system is proposed. Specifically, within a car park area, each parking lot is equipped with one inexpensive sensor node, which detects the occupation or vacancy of this parking lot. A base station collects data relative to the status of all parking lots and sends periodic reports to a database. This database can be accessed by the upper layer management system in order to execute several tasks, such as vacant parking lots discovery, security supervision, automated toll, or/and statistic report creation.

In [48], a system to measure and classify road traffic based on WSN is proposed. The nodes of this system have integrated magnetic sensors and are deployed along the road in order to provide real time traffic monitoring. This configuration has been proposed as an easy and cheap to implement alternative to inductive coils that are traditionally deployed in roads for basic traffic control tasks. The architecture adopted in the system proposed is graphically presented in Figure 17.

In [49], a system for road traffic control near flooded tunnels is proposed. The specific system, consists of a WSN and a centralized control system. The analogue output signals of sensors, located in the interior of two underground tunnels, are converted to digital ones which are next transmitted through serial-to-Ethernet converters to an access point. Following a corresponding procedure the signals reach a programmable logic controller, which monitors the level of water in the interior of two

underground tunnels and accordingly not only regulates the water drainage of flooded tunnels by using pumps, but also performs traffic control of vehicles approaching the entrance to the tunnels.

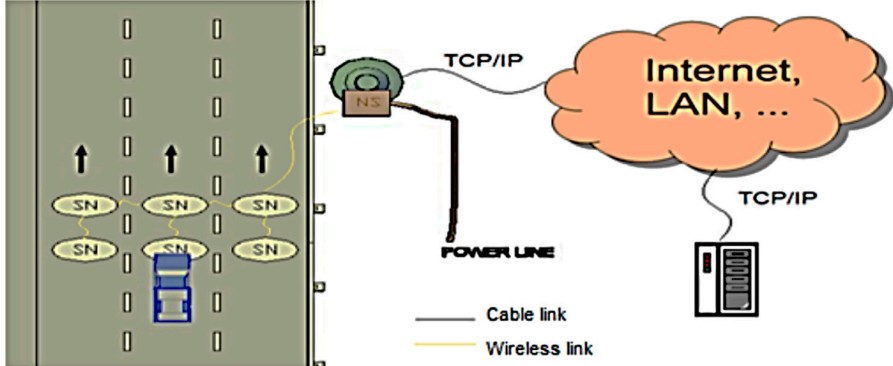

**Figure 17.** A depiction of the various setups where WSNs can be deployed [48].

A WSN based system for street lighting monitoring and control is presented in [50]. This system enables the remote control of street lighting lamps, by using Doppler sensors to allow for vehicle detection. The light intensity of the lamps is increased, to a preset level, in the presence of approaching vehicles and reduced in the absence of them.

### 2.6.2. Smart Homes

In the era of informatics, various systems can improve human lives. Wireless sensor networks can be applied in the indoor environment as in smart homes where machine to machine communications take place. Two typical examples are the indoor localization and motion monitoring and the monitoring of the indoor air quality.

Specifically, the monitoring of indoor air quality (IAQ), which is a term that refers to the air quality of a building, has become, with the rapid urbanization process that is taking place, one of the most important topics of WSN urban applications. Modern people spend a great part of their life within a building, thus the air quality plays a critical role for their health, safety, and comfort. In order to monitor IAQ the authors of [51] developed a WSN application that utilizes wireless nodes equipped with gas sensors for sensing and communicating wirelessly in real time.

Similarly, the WSN based IAQ monitoring system, which is described in [52], uses sensor nodes that measure temperature, relative humidity, and concentration of carbon dioxide in classrooms in order to correlate the level of indoor air quality with the students' level of performance in their studies.

Another popular application of indoor localization and motion monitoring that uses WSNs, is described in [53]. The authors developed a system, which is used to estimate the position of a person within an indoor environment. The system utilizes mobile sensor nodes worn by the moving persons and a network of static sensor nodes at known locations. To calculate the person's position, the system gathers data from the nodes in a central PC and applies Monte Carlo based estimation algorithms to track the moving persons in real time.

### 2.6.3. Transportation Systems

The transportation of vehicles within the urban environment is an important factor for the safety and proper function of public roads. In the modern era various researchers have proposed many systems in the transportation domain, which by utilizing WSNs can improve the safety of roads and provide information during the use of them by drivers. Below we present some relative WSN applications applied in the transportation domain.

In [54] the authors describe a WSN application for VANETs. This application using Secure Multimedia Broadcast Framework (SMBP) is to monitor a persons' way of driving by utilizing a

processing unit on the vehicle, which analyzes data, collected from the sensors in order to calculate the cost of the driver's insurance. The described system is utilized in order to determine how much the driver detects nearby vehicles, drives with caution, follows the traffic law, or uses a mobile device. The sensing system includes a static sensor board that provides the sensed data via cellular network to the control center and a Global Position System/Universal Mobile Telecommunications System (GPS/UMTS) module in order to connect to the Internet. The basic configuration of SMBP is depicted in Figure 18.

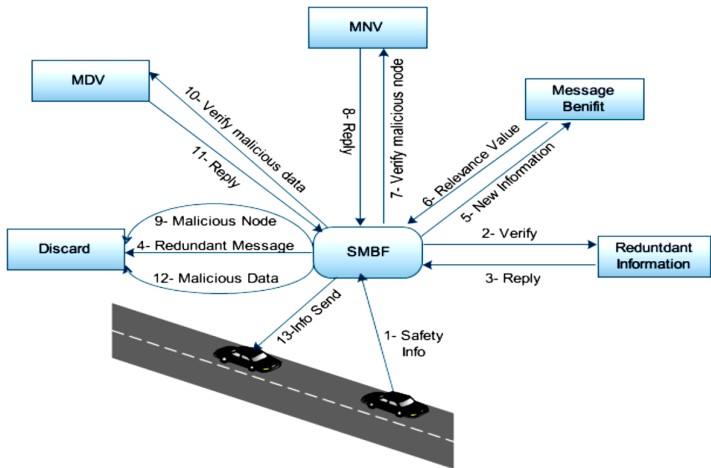

**Figure 18.** A representation of the Secure Multimedia Broadcast Framework (SMBP) [54].

In [55] a system is proposed, which analyzes the performance of a vehicle on the road by utilizing a telematics WSN. The system utilizes deployed sensor nodes on the road and a sink node on the roadside. The sensors nodes detect the passing vehicles and transmit data to the sink node. Furthermore, when an event occurs a packet is created, the sink node implements a timestamp to it and forwards it to the com node, which utilizes a level-based static routing protocol. Next, the com node forwards this packet to the user, which analyzes the provided information in order to determine various metrics of the road such as a vehicle's speed, lane vehicle density etc. A depiction of the overall system configuration is shown in Figure 19.

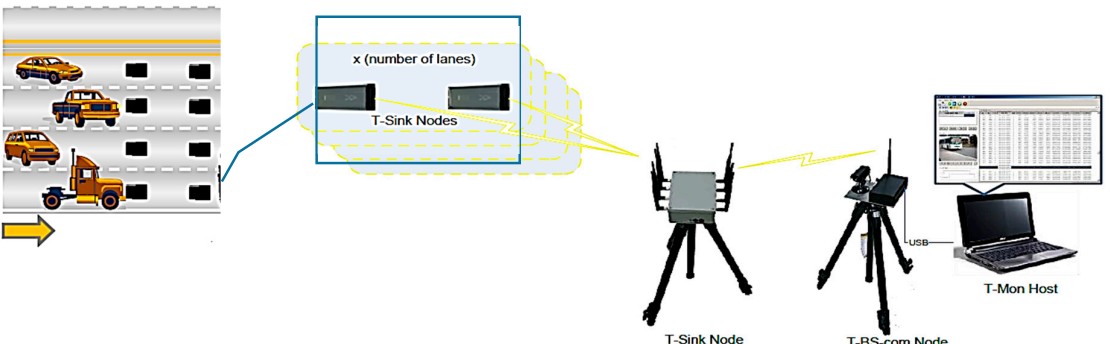

**Figure 19.** A graphical depiction of the WSN based transportation monitoring system [55].

In recent years, various WSN have been deployed within cities resulting in the formation of small WSN based islands. In [56] the authors describe an integration of these existing islands deployed along the roadside, in order to be utilized for post-accident management and to prevent accidents. Furthermore, the authors utilize two wireless standards i.e., IEEE 802.15.4 and IEEE 802.11p due to the need for separate roadside sensor units. In order to prevent accidents, the WSN islands constantly detect road conditions and transmit information to the vehicles passing by the area, which aggregate

these data. Then the vehicle that has this information, transmits warning messages to nearby vehicles in order for the data to be disseminated along the road. The post-accident gathered information is restricted, by the authors, in order to be used only by authorized personnel such as insurance companies, criminological teams, or other authorities. Moreover, according to the authors, the post-accident management can function properly by communicating with the WSN islands on the roadside, without the use of VANETs. The overall system operation is illustrated in Figure 20.

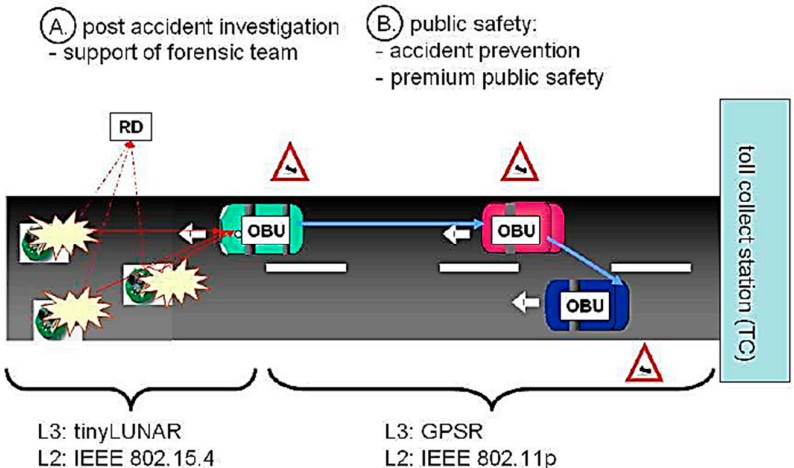

**Figure 20.** An illustration of the WSN deployed for an Intelligent Transport system [56].

The researchers of [57], presented a mechanism that controls the speed of a vehicle by utilizing an infrastructure-to-vehicle communication via WSNs. In the described system the traffic lights utilize long-range RFID tags, which can measure a vehicle's speed and communicate with the vehicle's onboard RFID tags, to avoid collisions. The onboard RFID tags are able to provide feedback to actuators, which can change the longitudinal speed by controlling various parameters of the vehicle such as braking and throttling. Furthermore, the authors created a fuzzy logic algorithm enabling the automatic control of the vehicle. Structural elements of the system developed are shown in Figure 21.

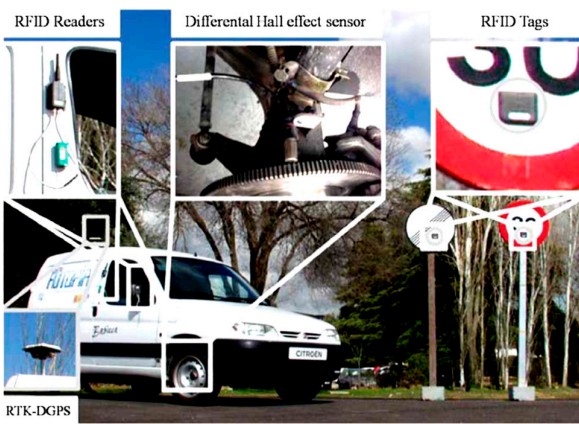

**Figure 21.** A depiction of the various sensors deployed on a car for experimenting [57].

### 2.6.4. Structural Health Monitoring

The aim of Structural Health Monitoring (SHM) in buildings and other types of civil engineering substructures, is to monitor their integrity and to detect the existence and the extent of damages in the materials or/and the structure of their bodies. The use of wireless sensors facilitates the accomplishment of tasks of this kind both in a periodic basis and after critical events (e.g., earthquakes).

A WSN application of this kind is presented in [58]. Specifically, a WSN incorporating nodes with appropriate sensors, such as accelerometers and strain gauges, was developed to monitor the restoration works carried out in an old church building located in Italy, which was damaged after an earthquake. The system monitored the on the fly response of the structure to vibrations and enabled the transmission of alert signals when needed.

In [59] the design, development, and deployment of a, WSN based, structural health monitoring platform for a stadium located in the USA is described. By using vibration sensing, the system collects real time, data during athletic and other major events to verify the structural behavior of the stadium, in correlation with the behavior of the audience.

Another structural health monitoring system which makes use of a WSN is presented in [60]. This system is designed, implemented, deployed, and tested to monitor the structural performance of a bridge in China by sensing vibration signals that are produced under various conditions in specific points of interest located at the body of the bridge.

## 3. Discussion

There is no doubt that WSNs enjoy remarkable capabilities which make them ideal for an ever-extending range of applications. On the other hand, the operation of WSNs comes up with serious problems. Some of them are application dependent.

For instance, in military applications the required physical dimensions and weight of nodes are application dependent. For instance, in some surveillance applications the sensor nodes have to be extremely small in order to be undercover, while in many other military applications, physical dimensions and weight of the nodes are not considered to be important restrictions. On the other hand, nodes in such applications should definitely have an adequately extensive communication range (maybe ≥1 km), while the area to be covered is several square kilometers. Also, communication should attain optimal throughput, reliability, security, and resistance to jamming and intervention. Moreover, nodes should be robust enough to resist severe ambient conditions. Similarly, the WSN should be tolerant to the loss of a certain quantity of nodes.

In health applications, the physical dimensions and weight of the nodes have to be as small as possible particularly in the cases where they are wearable. Conversely, there is not any need for an extended communication range of nodes or area covered. Communication should be fault tolerant, fast, and reliable while jamming should be definitely avoided because the transmission of data is absolutely vital when time critical alert information is sent.

In flora and fauna applications, the limitations, if any, for physical dimensions and weight of the nodes are application dependent. For instance, in WSNs used in livestock farming, nodes which are implanted under the skin of animals must be as small as possible while in agricultural applications such limitations are usually absent. Also, in flora and fauna applications, nodes should be robust enough to stand ambient conditions. Similarly, the WSNs should be tolerant to the loss of a certain quantity of nodes. In addition, in some cases there is need for an extended communication range of nodes and area covered. The volume of data transferred is usually high, but the communication standards that should be met are not very high.

In environmental monitoring the physical dimensions and weight of the nodes are not considered to be the first requisite. Instead, the construction of the nodes has to be extremely robust in order to tolerate severe ambient conditions. Also, both the communication range and the area covered should be adequately wide. Additionally, communication should be resistant to jamming because emergency alerts should be transmitted without delay. Moreover, the WSN should be tolerant to the loss of a certain quantity of nodes.

In industrial applications most tasks are time critical while in the industrial environment the presence of electromechanical interference is remarkable. Therefore, communication should achieve optimal throughput, reliability, and resistance to jamming and interference. It may also be necessary to

operate under strict security standards. The communication range and the area covered depends on the nature of the specific application and as do the physical dimensions and weight of the nodes.

In the urban domain, there are different conditions that must be fulfilled in indoor and outdoor applications. Precisely, in WSNs for indoor use nodes have to be of relatively small physical dimensions and weight. Usually there is no need for the communication range and the area covered to be wide. Conversely, communication should attain high security to protect privacy and should resist interference caused by other home appliances. On the other hand, in outdoor urban applications, the physical dimensions and weight of nodes are of minor importance. Yet, both the communication range and the area covered need to be wide. Also, the communication should achieve high levels of throughput, security, reliability, and resistance to jamming and intervention due to the huge volume of data transmitted. Moreover, nodes should be robust enough to ambient conditions. Also, the WSN should be tolerant to the loss of a certain quantity of nodes.

The abovementioned considerations, regarding the required features that WSNs should have per type of application, are synoptically presented in Table 1.

**Table 1.** Required specifications of Wireless Sensor Networks (WSNs) per type of application.

| Type of Application | | Required Specifications | | | | | | |
|---|---|---|---|---|---|---|---|---|
| | | Node Weight and Dimensions | Node Robustness | Communication Range | Communication Throughput | Communication Reliability | Communication Security | Network Tolerance |
| Military | | Application dependent | Very High | Wide | Very High | Very High | Very High | Very High |
| Health | | Small | High | Small | Very High | Very High | High | High |
| Flora and Fauna | | Application dependent | High | Wide | Medium | Medium | Low | High |
| Environmental | | Of minor importance | Very High | Wide | Very High | High | High | Very High |
| Industrial | | Application dependent | Very High | Application dependent | Very High | Very High | High | Very High |
| Urban | Indoor | Small | Medium | Small | Medium | Very High | Very High | High |
| | Outdoor[M1] | Of minor importance | Very High | Wide | Very High | Very High | Very High | Very High |

Additionally, apart from the aforementioned issues that are application dependent, the operation of WSNs is also obstructed due to general issues, such as difficulties of wireless communication and weaknesses of the nodes.

Specifically, sensor nodes of WSNs suffer from extremely strict energy constraints. This is because their energy is typically supplied by batteries which are usually impractical to be either recharged or replaced, since the locations of sensor nodes are usually difficult or even impossible to reach. Therefore, the attainment of energy conservation is a vital issue for WSNs. For this reason, energy inefficiencies that exist at every one of the five layers of the protocol stack of sensor nodes must be eliminated. Given that data transmission is by far the most energy consuming task of nodes, power control schemes [61], data aggregation schemes that decrease the size of data transferred [62], [63] and energy efficient routing protocols [64] have been proposed. Likewise, in applications in which multimedia data are transmitted, the use of compression and restoration schemes provide a substantial reduction of communication load [65,66]. Additionally, the presence of excessive data traffic in a specific region of a WSN causes network congestion which obstructs data transmission, generates packet losses, and decreases the network throughput. For this reason, methodologies for congestion avoidance [67–69], congestion control [70,71], and load balancing [72,73] are used. Furthermore, each time that a node is disconnected from the rest of the network, due to, malfunction, damage, or energy depletion the communication for the remainder becomes more difficult and the communication cost is increased. For this reason, methods for the preservation of network connectivity are essential to be applied [74,75]. Moreover, it is anticipated that the nodes achieve the best exploitation of their sensing range and their communication range in order to cover as much of the network area. This is why coverage maximization methods

are considered [76,77]. Moreover, in WSN applications where multimedia data are transmitted, the attainment of high Quality of Service (QoS) is a necessity. Furthermore, the accomplishment of energy efficient routing in WSNs is often influenced by other issues such as the QoS attained [78,79]. Last but not least, in most WSN applications, the data transmitted within the network must be protected from any unauthorized use. For this reason, security preservation schemes are used [80,81].

## 4. Conclusions

The usage of WSNs already provides remarkable advantages for various domains of human activity. Thanks to the continuous evolution of technology both the capabilities of sensor nodes will keep expanding and their manufacturing costs will become lower. This is the reason why the range of WSN applications is expected to carry on growing.

In this article, the utilization of WSNs in specific domains, namely military, environmental, flora and fauna, health, industrial, and urban, was examined via the investigation of corresponding typical examples, both novel and well-known ones. From this examination, it became evident that the usage of WSNs not only provides numerous advantages in specific domains when compared against the relative means and methods that were traditionally used, but it also introduces novel applications. Additionally, for various applications both the problems and solutions developed, were identified and discussed.

The combinational utilization of relative methodologies and tools will assist both the enhancement of existing applications and the development of novel ones. On the other hand, certain problems that obstruct the usage of WSNs, such as energy limitations, congestion, connectivity loss, inadequate coverage, low QoS, and susceptible security, will remain at the center of scientific research.

**Author Contributions:** All authors have equally contributed to the work reported. All authors have read and agreed to the published version of the manuscript.

**Funding:** This research received no external funding.

**Conflicts of Interest:** The authors declare no conflict of interest.

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
