# Peer review of "Applications of Wireless Sensor Networks: An Up-to-Date Survey"

_asi, doi:10.3390/asi3010014_

Round 1

Reviewer 1 Report

The authors described the application areas for Wireless Sensor Network (WSN). However, there are some drawbacks for the authors to comment/answer,

1) Since the author want to discuss WSN and its applications, there is not a basic diaphragm/figure/table regarding to the formation of WSN, the different wireless communication protocols (regarding to communication standard, power consumption, distance range, network formation, etc). While the reviewer believe that will help readers to better understand the WSN and also the following part.

2). Besides, in each application/area, there is not one single figure existing in this manuscript. And figures should give readers a better understanding and illustration compared with only words.

3) Regarding to the introduction part, the logic should be further tuned. For example, in paragraph 3, "however, the inexpensive production of sensor nodes of this..." should be "Besides, the inexpensive production of ...". 

Author Response

The authors of the manuscript entitled "Up-to-date Applications of Wireless Sensor Networks: A Review" submitted to ASI Journal, wish to express their most sincere thanks to the Reviewers of this article for their valuable comments.

In what follows, each one of the comments made by Reviewer 1 is enlisted followed by the corresponding respond of the authors.

Comment 1:

 Since the author want to discuss WSN and its applications, there is not a basic diaphragm/figure/table regarding to the formation of WSN, the different wireless communication protocols (regarding to communication standard, power consumption, distance range, network formation, etc). While the reviewer believe that will help readers to better understand the WSN and also the following part.

Respond to Comment 1:

The introductory information given in Section 1 has been supported by the addition of Figure 1 and Figure 2 which illustrate the architecture of a wireless sensor node and a wireless sensor network respectively.

Comment 2:

Besides, in each application/area, there is not one single figure existing in this manuscript. And figures should give readers a better understanding and illustration compared with only words.

Respond to Comment 2:

Figure 3, which provides an illustrative overview of WSN applications has been added in Section 2. Moreover, Figures 4, 5, 6, 7, 8, and 9 have been added in Section 2 too, in order to depict each one of the main categories of WSN applications along with the corresponding sub-categories and the types of the sensors that are used in these applications. Furthermore, 2 new subcategories and 13 new applications have been added in order to enhance more the specific survey.  

Comment 3:

Regarding to the introduction part, the logic should be further tuned. For example, in paragraph 3, "however, the inexpensive production of sensor nodes of this..." should be "Besides, the inexpensive production of ...". 

Respond to Comment 3:

All recommended corrections have been made and the text has been extensively revised.

Reviewer 2 Report

My opinion is that the presented article is out of the scope for the journal. In the "Aims and Scopes" section of ASI journal it clearly says that "the journal is devoted to publishing research papers in the fields of integrated engineering and technology". Also, it states that "The main goal of this journal is to publish research results in applied system innovation. The ultimate aim is to discover new scientific knowledge relevant to designs of the future". (Reference: https://www.mdpi.com/journal/asi/about).

In this case, the authors present a review of the state of the art of Wireless Sensor Networks. Therefore, it is not a research paper and it does not discover any new scientific knowledge, i.e. it does not provide research results. In any case, if this review format is accepted in this journal, I still think that the authors should make an extra effort. First of all, I do not think that 68 references are enough to cover such a large topic on WSN. Also, the authors do not include a single figure throughout the paper. I think they should prepare at least one original figure as a summary of their review and also some other figures taken from the most important references they consider. The reading of the text without any supporting figure is quite tiresome and tedious.  

Apart from that, the authors should also take care of some grammar issues (e.g. "among of" is not correct) and formatting style (e.g. the use of italics for the sections is not coherent in the whole text). 

Author Response

The authors of the manuscript entitled "Up-to-date Applications of Wireless Sensor Networks: A Review" submitted to ASI Journal, wish to express their most sincere thanks to the Reviewers of this article for their valuable comments.

In what follows, each one of the comments made by Reviewer 2 is enlisted followed by the corresponding respond of the authors.

Comment 1:

First of all, I do not think that 68 references are enough to cover such a large topic on WSN.

Respond to Comment 1:

Following the reviewer’s recommendation, 2 new subcategories of WSN applications and 13 new applications have been added in order to enhance more the specific survey.  

Comment 2:

Also, the authors do not include a single figure throughout the paper. I think they should prepare at least one original figure as a summary of their review and also some other figures taken from the most important references they consider. The reading of the text without any supporting figure is quite tiresome and tedious.  

Respond to Comment 2:

The introductory information given in Section 1 has been supported by the addition of Figure 1 and Figure 2 which illustrate the architecture of a wireless sensor node and a wireless sensor network respectively. Also, Figure 3 which provides an illustrative overview of WSN applications has been added in Section 2. Moreover, Figures 4, 5, 6, 7, 8, and 9 have been added in Section 2 too, in order to depict each one of the main categories of WSN applications along with the corresponding sub-categories of applications and the types of the sensors that are used in these applications.

Comment 3:

Apart from that, the authors should also take care of some grammar issues (e.g. "among of" is not correct) and formatting style (e.g. the use of italics for the sections is not coherent in the whole text). 

Respond to Comment 3:

All recommended corrections have been made and the text has been extensively revised. The use of italics has been restricted only for the denotation of individual subsections, as requested by MDPI format regulations.

Round 2

Reviewer 1 Report

The quality of the manuscript is improved after the authors revision. In details, the presentation is improved and figures are also added. However, there still exists some questions to be solved before publication. 

1). Figures that discusses the concept of sensor/WSN are added. However, the quality of the figures should be improved (the font size and image resolution). Besides, since the authors cited a lot of references to support the WSN application areas, it is better to add some applications figures (e.g., some existing sensors/networks). 

2). A lot of different wireless technologies have been introduced in this review. A table that describes the pros and cons of these technologies should be included for better presentation, which will also benefit readers.

3). The presentation should also be improved. For example, in the discussion part, some paragraphs only have one/two sentences. 

Author Response

The authors of the manuscript entitled "Up-to-date Applications of Wireless Sensor Networks: A Review" submitted to ASI Journal, wish to express their most sincere thanks to the Reviewers of this article for their valuable comments.

In what follows, each one of the comments made by Reviewer 1 is enlisted followed by the corresponding respond of the authors.

Comment 1:

Figures that discusses the concept of sensor/WSN are added. However, the quality of the figures should be improved (the font size and image resolution). Besides, since the authors cited a lot of references to support the WSN application areas, it is better to add some applications figures (e.g., some existing sensors/networks). 

The authors of the manuscript entitled "Up-to-date Applications of Wireless Sensor Networks: A Review" submitted to ASI Journal, wish to express their most sincere thanks to the Reviewers of this article for their valuable comments.

In what follows, each one of the comments made by Reviewer 1 is enlisted followed by the corresponding respond of the authors.

Comment 1:

Figures that discusses the concept of sensor/WSN are added. However, the quality of the figures should be improved (the font size and image resolution). Besides, since the authors cited a lot of references to support the WSN application areas, it is better to add some applications figures (e.g., some existing sensors/networks). 

Respond to Comment 1:

Following the reviewer’s recommendation, the 9 figures existing in the previous version were redrawn in order to attain higher quality of presentation. Moreover, 15 new figures were added in order to support the description of the applications presented.

Comment 2:

A lot of different wireless technologies have been introduced in this review. A table that describes the pros and cons of these technologies should be included for better presentation, which will also benefit readers.

Respond to Comment 2:

Following the reviewer’s recommendation, a table discussing the different requirements of the various types of applications regarding 7 structural and operational specifications was added in Section 3, in order to augment the comprehension of the findings.

Comment 3:

The presentation should also be improved. For example, in the discussion part, some paragraphs only have one/two sentences. 

Respond to Comment 3:

Following the reviewer’s recommendation, the text has been extensively revised and the discussion section was rewritten according to the reviewer’s recommendations.

Reviewer 2 Report

Although some of my comments have been addressed (e.g., the inclusion of more references), I think that the issue with the figures is still there. When I suggested the inclusion of more figures I was not talking about simple diagrams drawn using Word... I do not consider that the overall quality of this review is good enough for its publication. The authors should check any published review to have an idea of the type of figures that they should include. 

Author Response

The authors of the manuscript entitled "Up-to-date Applications of Wireless Sensor Networks: A Review" submitted to ASI Journal, wish to express their most sincere thanks to the Reviewers of this article for their valuable comments.

Comment :

Although some of my comments have been addressed (e.g., the inclusion of more references), I think that the issue with the figures is still there. When I suggested the inclusion of more figures I was not talking about simple diagrams drawn using Word... I do not consider that the overall quality of this review is good enough for its publication. The authors should check any published review to have an idea of the type of figures that they should include.

Respond to Comment :

Following the reviewer’s recommendation, 51 new figures were added in order to support the description of the applications presented.  The software used by the authors, for the design of their own figures is draw.io. Moreover, one table that discusses the different requirements of the various types of applications regarding 7 structural and operational specifications was added in Section 3. 

Round 3

Reviewer 1 Report

The authors have carefully revised the manuscript and have answered questions the reviewer raised. And I recommend "Accept in present form" for this manuscript.

Reviewer 2 Report

All my comments have been addressed.